# Waves and currents decrease the available space in a salmon cage

**Ása Johannesen**[1]*, **Øystein Patursson**[2], **Jóhannus Kristmundsson**[3], **Signar Pætursonur Dam**[4], **Mats Mulelid**[5], **Pascal Klebert**[5]

1 Vistfrøði, Fiskaaling, Hvalvík, Faroe Islands, 2 RAO, Kirkjubø, Faroe Islands, 3 Fjarðaelvi, Fiskaaling, Hvalvík, Faroe Islands, 4 Hiddenfjord, Sørvágur, Faroe Islands, 5 Sintef OCEAN, Trondheim, Norway

* asajoh@fiskaaling.fo

## Abstract

Due to increasing demand for salmon and environmental barriers preventing expansion in established sites, salmon farmers seek to move or expand their production to more exposed sites. In this study we investigate the effects of strong currents and waves on the behaviour of salmon and how they choose to use the space available to them. Observations are carried out in a site with strong tidal currents and well mixed water. Using video cameras and echo sounders, we show that salmon prefer to use the entire water column, narrowing their range only as a response to cage deformation, waves, or daylight. Conversely, salmon show strong horizontal preference, mostly occupying the portions of the cage exposed to currents. Additionally, waves cause salmon to disperse from the exposed side of the cage to the more sheltered side. Even when strong currents decrease the amount of available space, salmon choose to occupy the more exposed part of the cage. This indicates that at least with good water exchange, the high density caused by limited vertical space is not so aversive that salmon choose to move to less desirable areas of the cage. However, the dispersal throughout the entire available water column indicates that ensuring enough vertical space, even in strong currents, would be beneficial to salmon welfare.

## Introduction

Aquaculture is a major provider of fin fish protein consumed globally, accounting for approximately 52% of all fish produced for human consumption [1]. In aquaculture, Atlantic Salmon account for only 4.5% of production in weight, but 19% in value. While aquaculture production is increasing globally, salmon production in the Atlantic is stagnating. The causes mainly relate to complete exploitation of available farming sites, with pollution and parasite infestations being the major factors limiting expansion in near-shore sites [2].

Salmon lice (*Lepeophtheirus salmonis*) are a major parasite in Atlantic salmon farming. They spend their parasitic life stages on the salmon where they consume mucous, blood, and skin, which leads to sores and in extreme cases, mortality [3, 4]. Even sub-lethally, salmon lice are a cause of poor welfare due to the stress and pain caused both by the parasites themselves and the removal methods [5, 6].

**Data Availability Statement:** Scripts and raw data files are available at: 10.5281/zenodo.5974484.

**Funding:** The Norwegian Research Council: 267800/E40 (FUTUREWELFARE) and Centre for Research-Based Innovation on exposed aquaculture operations (SFI EXPOSED):237790/O30 supported this work in grants to Pascal Klebert and Ása Johannesen respectively. Fiskaaling also supported this work. The funders had no role in study design, data collection and analysis, decision to publish, or preparation of the manuscript.

**Competing interests:** The authors have declared that no competing interests exist.

Because of the ever escalating challenges posed by salmon lice and the limitations put on biomass in near shore sites, the salmon aquaculture industry is making investments in adapting current farming methods. One such adaptation is to move farms out to more exposed locations where higher water exchange can mitigate pollution and possibly dilute infectious sea lice [7–9]. At the current rate of development, salmon will experience substantially larger waves in substantially different farm constructions than those being used in industry for the last 20 years [10].

Due to the forthcoming changes in salmon aquaculture, much work has been carried out in order to determine how well salmon are able to cope with the currently most extreme conditions [9]. From these studies, some information is available on swimming speed capacity [11, 12], sensitivity to variation in temperature and oxygen saturation [13–16], swimming energy expenditure, and how a variety of these factors affect growth and feed conversion [17–19].

The behaviour of salmon in relation to currents and time of day has also been extensively studied. Salmon change their swimming behaviour according to the availability of light in their environment. When lights are deployed at night during winter, salmon will maintain daytime swimming behaviours and navigate to the depth of the lights [20], while they disperse more and decrease swimming activity if no lights are deployed [21]. This behaviours is modified by other environmental factors, such as water temperature [22]. Current speed affects swimming behaviour in two major ways. Strong current decreases aggressive interactions and increases shoal cohesion in raceways [11] and in salmon cages a strong current changes swimming mode from circular to standing on current, that is maintaining position in the cage while swimming against the current [23].

Currents do not only affect the swimming behaviour and energy expenditure of salmon, they also affect the shape of salmon cages. Cage deformation due to current can cause a decrease in cage volume of 20% at 0.5 m s$^{-1}$ [24, 25]. While not as thoroughly studied, there are indications that waves too can change the shape of a salmon cage [26].

There is some recent work on how behaviour is affected by waves, mostly investigating vertical preference and swimming effort [27, 28]. However, while there is data on how changes in the available space within a cage affect biomass and salmon welfare [29], the effects of waves on available space and salmon behaviour have not been thoroughly investigated.

In addition to increasing demand for salmon driving industry to innovate, there is also an increased awareness of fish welfare considerations [30, 31]. This means that there is growing consumer pressure for not only environmental certifications such as ASC (Aquaculture Stewardship Council) [32], but also for assurances that the farms can deliver a minimum welfare standard, such as the RSPCA assured scheme [33]. Being able to farm fish in exposed locations without compromising on welfare requires extensive knowledge of how these new conditions will affect the fish. While most salmon farmers have intimate knowledge of already established sites, it is still necessary to be able to generalise such knowledge to new and more exposed sites. This study is an attempt at detailing how salmon are affected by the combined effects of waves and currents. Particularly, how the hydrodynamic conditions change their preferred positioning in the cage and how they change their behaviour.

Here, we monitor the behaviour of salmon and how they use the space available to them in a cage exposed to both currents and waves throughout the winter months of 2019/2020.

We predict that currents cause salmon to move upwards in the water column, and that this is at least partially caused by the cage bottom being pushed upwards by the current.

We predict that daylight and waves will cause the salmon to move downwards in the water column, and that situations where large waves and strong currents coincide is where salmon have the most limited space available to them.

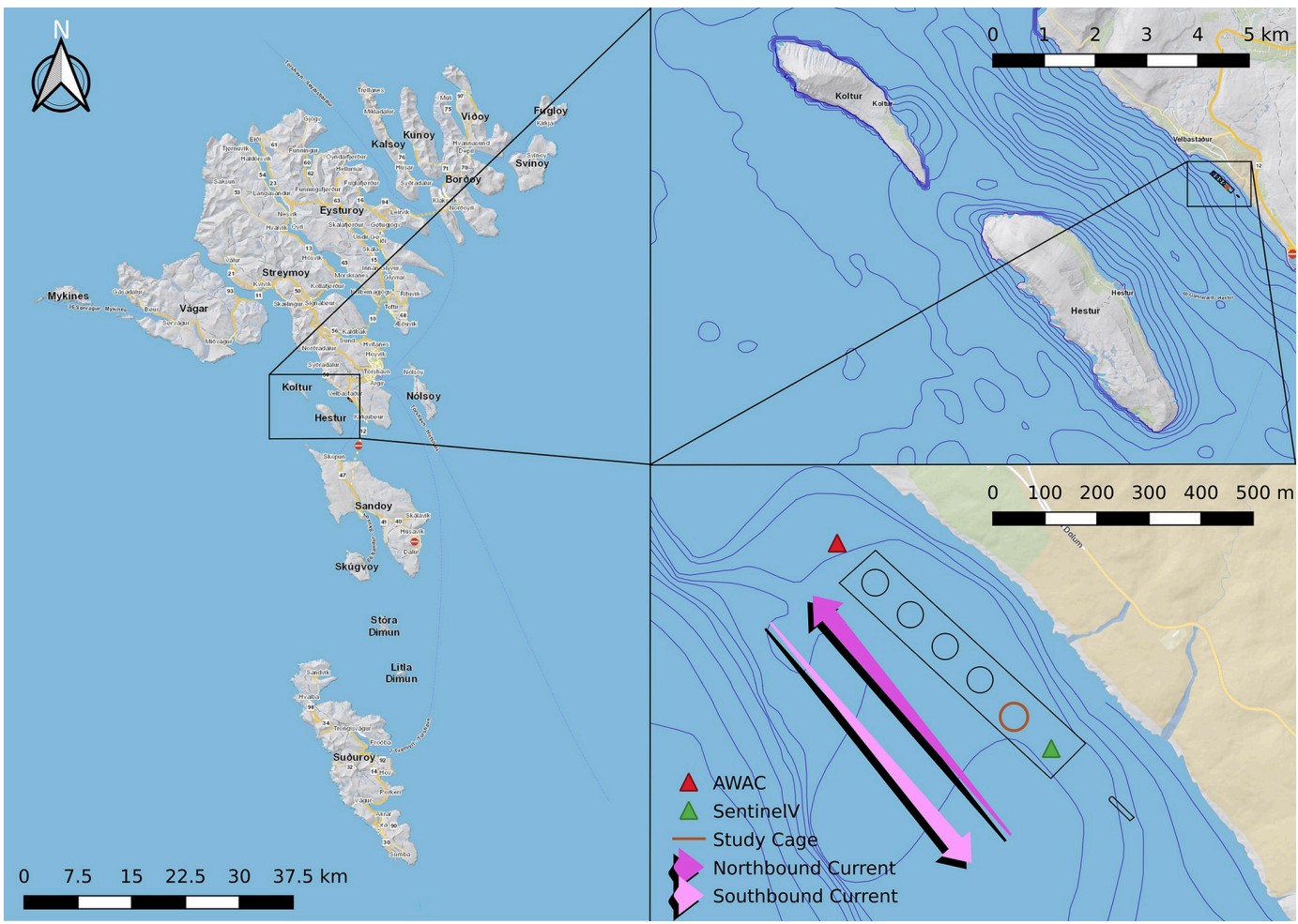

**Fig 1. Study site.** The map shows precise close up location of the farm and study cage with 5 m depth contours as well as zoomed out maps of general location in relation to surrounding islands with 10 m depth contours. Map reprinted from kort.foroyakort.fo under a CC BY license, with permission from the Faroese Environment Agency, Umhv∅rvisstovan. Original copyright 2022.

## Materials and methods

### Field site

Work was carried out at Hiddenfjord's "Velbastaður" salmon farm on the Faroe Islands, which is a site that is exposed to strong tidal currents as well as large waves in winter. The cages are arranged in a straight line along the coast, which is oriented close to a Northwest-Southeast axis (Fig 1). A small island ("Hestur") west of the farm creates a strait through which tidal currents move, alternating between northbound and southbound currents. The location of the farm is quite shallow (25-30m depth) and the tidal currents ensure good water exchange and a well mixed water column. Due to the hydrodynamic conditions at the site, the side of the farm nearer the shore is more sheltered than the outer side. Depending on wind direction, waves enter the strait either from the south or north of Hestur.

For the purposes of this study, the southernmost cage was selected for observation. Being located at one end of the row of cages, it was highly exposed to currents, especially from the south. Additionally, the cage was expected to be exposed to waves entering the strait from

either side of Hestur. Ideally, several cages would be included from different sites in order to get a general picture of the relationship between waves, currents, and salmon. However, for reasons of logistics, this was not possible with such an extensive setup. Therefore, this study details the conditions in one cage that has been thoroughly monitored.

The cage was stocked with 112 thousand salmon with a mean body length of 54.0 ± 4.1 cm, weighing 2.16 ± 0.6 kg, which amounted to an approximate biomass of 15 kg m$^{-3}$. Additionally, the cage housed approximately 10 thousand lumpfish (*Cyclopterus lumpus*). The salmon were fed using air driven surface feed spreaders regulated using a feed camera, which detected uneaten pellets.

## Equipment setup

To monitor wave height and period and current speed and direction, two Acoustic Doppler Current Profilers (ADCPs) were deployed; one AWAC directly north of the northernmost cage and one Sentinel V directly south of the study cage (Table 1, Fig 1). Both profilers were set up to collect hourly wave data from a 20 minute burst. The Sentinel V measured current data at one ping every two seconds and the AWAC measured currents in bursts of 120 pings at one ping per second every five minutes.

Directly attached to the cage were several different sensors. The layout is presented in Fig 2.

To monitor vertical distribution of fish within the cage, two echo sounders (Table 1) were attached to the cage bottom, both positioned half way between the centre and side of the cage at opposite sides parallel to the coast line. The echo sounders were suspended from the bottom of the cage looking up, thereby recording distance to the surface as well as any fish within the echo sounder beam. This allowed us to measure the depth of the cage as well as which parts of the water column were occupied by fish. The echo sounders were set up to ping once every four seconds for the duration of the experiment. Lumpfish do not have swim bladders, so they do not show up clearly in the echo sounder. However, salmon cages on the Faroe Islands often contain a small amount of saithe (*Pollachius virens*), which have entered the cage when they were small enough to get through the net. These do have swim bladders, so the data from echo sounders include both salmon and saithe.

The echo sounders were equipped with sensors measuring tilt and pressure (Table 1). These were used to detect any instances where deviations from vertical would invalidate distance data and to validate the distance to the surface measured by the echo sounders. Tilt sensors recorded tilt and pressure every five seconds.

In addition to the echo sounders, six pressure sensors (Table 1) were attached to the cage in order to properly account for any cage deformation that may occur. These were set up to record pressure every five seconds.

**Table 1. Equipment deployed.**

| Type | Manufacturer | Specifications |
|------|--------------|----------------|
| *ADCP* | Teledyne | RDI Sentinel V 500 kHz with 50m range, wave and current |
| *ADCP* | Nortek | AWAC 600 kHz with 50m range, wave and current |
| *Echo sounder* | Simrad | EK 15 200 kHz 26˚ viewing angle |
| *Tilt sensor* | Star-Oddi | Starmon Tilt; tilt, pressure, and temperature logger calibrated to 50m |
| *Pressure sensor* | RBR | RBRSolo 3 D; Depth logger, range 1700m, calibrated to 50m |
| *Video camera* | JT Electric | 95˚ viewing angle, 1,3" sensor, and minimum 0.05 lux light sensitivity |

Location of equipment in the cage can be seen in Fig 2.

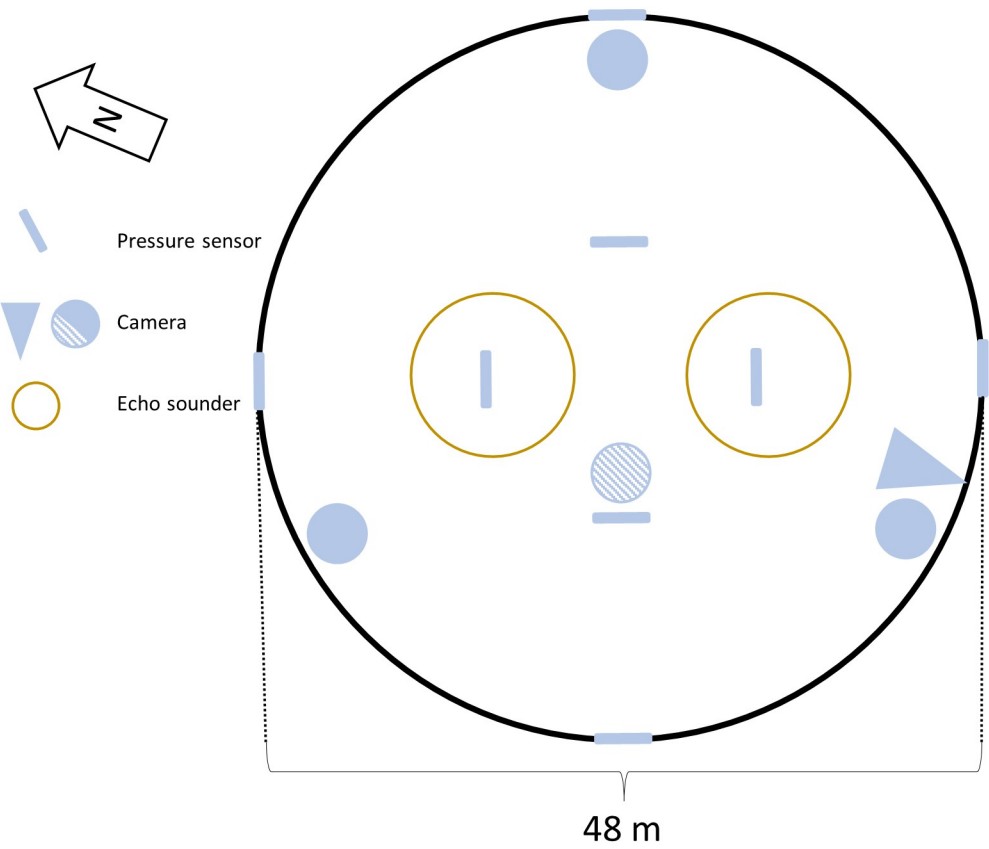

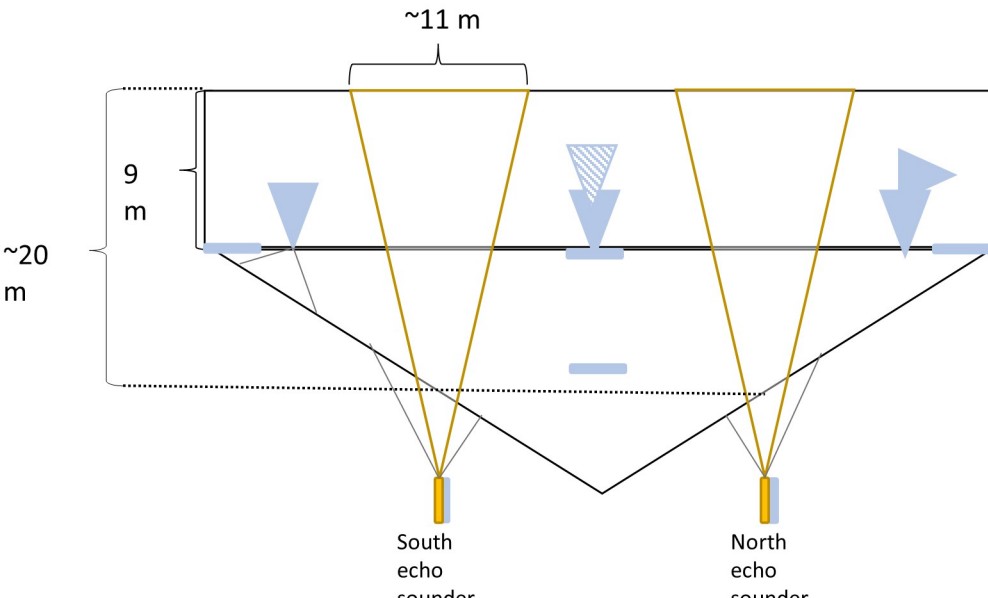

**Fig 2. Layout of equipment.** Aerial and side on view of the study cage with the location of echo sounders (yellow circles and cones show viewing area of echo sounders), pressure sensors (attached to the net and shown as blue bars) and video cameras (shown as circles or cones depending on viewing angle). Camera viewing angles are approximate. The camera with diagonal stripes is the feed camera, which was nearer the surface than the bottom mounted cameras.

To monitor salmon behaviour, five video cameras were positioned in the cage (Table 1). The camera that the salmon farmers used for monitoring feed consumption ("Feed camera") was included in our recording setup and was located approximately in the centre of the cage and nearer the surface (at 7 m depth) than our other upward facing cameras. Three upward facing cameras were located approximately equidistantly around the edge of the cage, one towards the south where northbound tidal currents entered the cage ("South"), one towards the north where southbound tidal currents entered the cage ("North") and one towards the east, which was the most sheltered location in the cage ("East"). The cameras were attached to the bottom of the cage and held up using a buoy (at 9 m depth). One camera was placed on the side of the cage at five metres depth looking inward ("Inwards") near the "South" camera. Camera "North" and "South" were positioned such that they would capture current related swimming behaviour, such as standing on current as well as any potential changes caused by waves. The more sheltered camera was used to record salmon avoiding currents or waves and whether consistent swimming behaviour (for example circling the cage) persisted throughout the cage. The camera looking inwards was used to capture close up video of salmon either orienting towards the current or swimming alongside the edge of the cage as well as capturing salmon near the surface of the water. Finally, the feed camera was used to capture presence within the centre of the cage outside of feeding times. Details about the instruments can be found in Table 1.

## Data collection

Data collection was carried out from the first of November 2019 to the 15th of February 2020 in order to capture as much as possible of both bad weather events as well as different tidal conditions.

Video cameras were remotely controlled using iSpy [34] and scheduled to record for five minutes each once per week. In addition to these baseline recordings, alternative schedules were enacted for bad weather events to record for five minutes three to four times each day during bad weather to capture behaviours in large wave conditions. At the end of the trial, the recorded wave data were sorted by wave height and period length and videos representing the full range were selected for analysis. Similarly, videos from days with weak current and strong current were also chosen for analysis, particularly from days where several videos were recorded within a day in order to capture the tidal current. This selection process resulted in 22 baseline videos in addition to 143 videos chosen to represent different hydrodynamic conditions. Unfortunately, when the weather was very bad, power did sometimes cut out, so it was not possible to assemble a perfectly balanced number of videos for all conditions. For swimming effort, most of the videos selected for behavioural observations were used, but in some videos, no fish were present, so these two data sets are not the same size.

Echo data were recorded continuously on a local hard drive and then uploaded remotely to a cloud server.

Current and wave data was recorded locally and downloaded when the ADCPs were recovered.

Tilt and pressure sensors stored data locally and data were downloaded when the sensors were recovered.

## Welfare monitoring

In order to ascertain the general welfare of salmon on the farm during the study period, we carried out Operational Welfare Indicator registrations. During the sampling period, Operational Welfare Indicators (OWIs) were recorded every two weeks when weather allowed

**Table 2. Operational welfare scores recorded.**

| Type | Fins | Pupil | Sclera | Snout | Skin |
|---|---|---|---|---|---|
| 0 | A little wear | Black | White | No injury | No injury |
| 1 | Damaged | White spot | 10% black | Slightly worn | < 2.5 cm |
| 2 | Missing fins | Big white spot | 25% black | Wound | > 2.5 cm |
| 3 | | Bleeding | > 50% black | | Bleeding |

Each salmon was scored on their overall condition, so both eyes and all fins were checked and the greatest score recorded for each.

(Table 2). Prior to this, welfare indicators were collected on a more ad-hoc basis to establish a baseline, and finally again at harvest. The large gap in data from March until harvest is due to the COVID-19 pandemic preventing fieldwork. The OWIs were adapted by Hiddenfjord from SWIM 1 and 2 [35, 36] to make them more practical to use as part of regular farm management practices. The OWIs were collected from 10 salmon from each cage in connection with routine louse counting. The salmon were caught using a dip net, anaesthetised in 60 mg L$^{-1}$ Finquel (MS-222) and lice numbers and gill condition were recorded before OWIs were recorded. After regaining consciousness in fresh seawater, the salmon were released back into the cages. For the purposes of this study, a sum of scores for each salmon excluding sclera colour was used to determine the overall welfare of the salmon. Sclera colour or eye darkening was not included here because while there is evidence that they can be used as an indicator of stress [37], this has not yet been well established in salmon. The total score excluding sclera can range from 0 to 10 with a low number indicating good welfare and a high number indicating poor welfare. Many individual based indicators in SWIM 1 and 2 are not included because there was either no variation in them (e.g. sexual maturation did not occur) or they did not relate to conditions on the farm at the time (e.g. deformities).

## Behavioural observations

Due to the nature of salmon moving in and out of camera, no attempt at counting the salmon was made. Instead, videos were coded in one of three qualitative categories throughout; "No salmon" (less than five salmon visible), "Some salmon" (more than five and less than 50 salmon visible), and "Many salmon" (More than 50 salmon visible). Finally, the presence of salmon near the surface was recorded for reasons of validating the near surface echo sounder data. When salmon were recorded in cameras with a view of the net, collisions with the net were also recorded.

In addition to the presence of salmon, the general behaviour of the salmon was recorded. Again here, we did not use a focal animal or try to record the behaviour of every animal. Instead, when more than 80% of the salmon performed the same behaviour (such as shoaling, standing on current, or directional swimming), that is what was recorded for the duration. When there was no clear majority behaviour in frame, no behaviour was recorded. Typically, this meant that the salmon were swimming in no coherent shoal with some salmon standing on the current and some salmon swimming in one direction and some other swimming in another direction. All video segments not classified as "standing on current" were classified as "free swimming" and all video segments where salmon were not together in a coherent shoal were classified as "not shoaling".

Finally, swimming effort was recorded by noting the time taken to beat with the tail fin three times by three different fish in each video.

## Data processing

Behaviours were extracted from videos manually using BORIS [38] and swimming effort was recorded using VLC [39] with the "Time" extension [40].

Echo sounder data were extracted from the raw data files using the "oce" package in R [41] and exported for further processing in Python [42]. First, the water surface was found. Second, data above the surface was removed, and depth adjusted to the surface rather than the distance from the echo sounder. Third, the data were binned into 5 minute intervals and 16.5 cm depths and $S_v$ (acoustic back scattering strength) averages were exported. Once data had been exported, the exported files were read into R, where the surface and the lowest 4.5 metres (below the cage bottom) were removed. The code for the echo sounder data processing can be found on Github [43].

## Data analysis

Analysis was carried out in R [44] and tidyverse [45] using the packages lubridate [46] and circular [47] for data cleanup, lme4 [48] and lmerTest [49] for statistical inference, performance [50] for model fit assessment, and ggplot2 [51] with colorspace [52] was used for plotting. All data and code used for the analysis is available on Github [43].

For video data, the following methods were used; to analyse swimming mode, a general linearised mixed effects model was used with a binomial (log link) family where each video was classified as salmon being mostly in either swimming mode, with current strength and current direction as predictors, and with camera as the random intercept term. The effects of environmental conditions on swimming effort was analysed using a linear mixed effects model with tail beats per second as the dependent variable, current speed, current direction, and wave period as predictors as well as camera as random intercept term. The reason for including camera as random intercept term in these models is that hydrodynamic conditions are not uniform throughout the cage, so the salmon in the different cameras will be affected differently by the conditions measured outside of the cage. The models described are the minimal adequate models, where variables that did not significantly affect the fit of the model have been removed. The amount of time where "Many" salmon appear in a camera (proxy for proximity to sides and surface) was analysed using a general linearised mixed effects model with time where "Many" salmon were visible in the camera as dependent variable, wave height (Hm0) and period (Tp) as predictors, and cameras as random intercept term. To assess the effect of hydrodynamic conditions on salmon collisions with the cage net, a poisson family general linearised model was used. The dependent variable was the number of collisions divided by the amount of time where there were salmon visible, to get a measure of collisions per second, multiplied by 300 to extrapolate to a 300 second (five minutes) long observation, and then rounded to whole number in order to use a poisson family model. Predictors were current speed and wave height. Shoal cohesion was investigated using a binary classification of videos to "shoaling" and "not shoaling" in a binomial family glm due to the highly bimodal nature of this variable where salmon were either mostly shoaling or not at all. Predictors were wave height, current direction and camera. Model fit for each model was assessed using diagnostic plots.

For echo data, linear models were used to estimate the effects of environmental variables on the "evenness" of fish dispersal within the water column using raw $S_v$ as a proxy for relative fish density. The residuals from these models indicated how variable $S_v$ was, so large residuals indicated "clumping" and small residuals indicated evenly dispersed fish. We also used linear models to how estimate how environmental variables affected fish swimming depth, weighing the depth variable by $S_v$. While it is possible to estimate real biomass from the back scatter

[53], we did not calibrate gain to do this, as we were more interested in relative changes rather than biomass estimation.

### Ethics statement

This study was not a manipulative experiment. However, we did install video cameras within the cage as well as echo sounders underneath it. We do not have reason to believe that these affected the salmon, but they were a deviation from the regular routines on the farm. We also handled the salmon in order to perform OWI monitoring, but this was already part of the management routine at Hiddenfjord farms, so didn't deviate from normal practices. Regardless, ethical approval was still applied for and given by Fiskaaling's Ethical Board (Approval number 007).

## Results

### Environmental conditions

During the monitoring period, currents at the north of the farm ranged from 0.00 to 0.49 m s$^{-1}$ with a mean current of 0.10 m s$^{-1}$. At the south of the farm, conditions were similar with a range between 0.06 and 0.51 m s$^{-1}$, and with stronger currents on average (0.16 m s$^{-1}$) (Fig 3). Flow direction was mostly bimodal switching between a north-westerly current and a south-easterly current, hereafter referred to as Northbound and Southbound current (Fig 3). Due to the difference in particularly current direction between the two ADCPs, the Sentinel V or a tidal analysis built on the data from the Sentinel V were used for further data analysis using current as a predictor.

The maximum significant wave height (Hm0) measured north of the farm was 3.02 m. South of the farm it measured (Hs) 3.24 m (Fig 4). Wave period (Tp) ranged from 2.07 to 22.55 seconds. Low and high wave heights were recorded at the same time in both ADCPs, indicating that they were exposed to similar wave heights (Fig 3). The wave data from the AWAC was used in analyses going forward due to the longer measurement period.

### Cage deformation

The bottom of the cage moved up from almost -20 m up to -6 m depth. The sides of the cage did not move as much, varying approximately three metres in depth between -9 m and -6 m (Fig 5). The direction of current affected how the pressure sensors moved with oncoming current causing the bottom of the net to move up farther than current from the lee side of the cage. In other words, in a northbound current, the bottom of the south side of the cage moved up more whereas in a southbound current, the bottom of the north side of the cage moved up more. The southern side of the cage was particularly affected in a northbound current, as there were no cages south of the study cage to shelter it.

### Video observations

**Swimming mode.** There was a connection between current speed and swimming mode. The number of videos where the most prevalent swimming mode observed was standing on current increased with current speed. This effect was decreased in southbound current (Current speed; z = 3.643, residual df = 151, P < 0.001, Flow direction; z = -3.683, residual df = 151, P < 0.001, Fig 6), which is consistent with the indication from the pressure sensors that northbound current affected the cage more than southbound current.

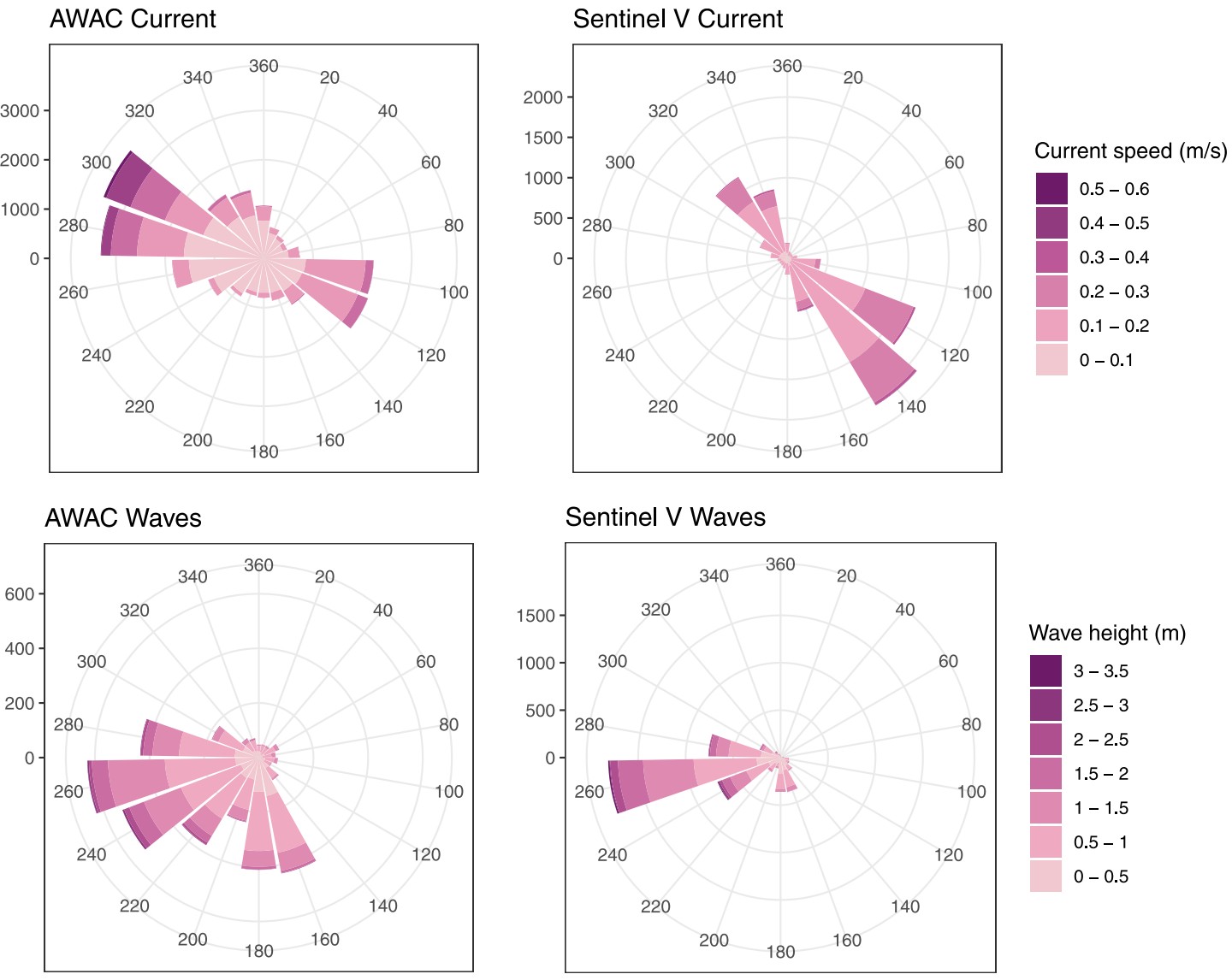

**Fig 3. Current and wave directions from two current profilers.** Wind roses of the currents and waves measured by the two profilers. Currents are towards the indicated directions, waves travel from the indicated directions.

**Swimming effort.** Because salmon recorded for swimming effort were more likely to be fish that spent a longer time in frame than those who spent a short time in frame, there are likely to be differences between these fish and the general population within the cage.

Swimming mode significantly affected tail beats per second with salmon beating faster with their tails when not swimming against the current ($F_{1,445} = 28.34$, P < 0.001). Because of this change in swimming effort related to swimming mode, data were analysed with swimming mode as a random predictor in a linear mixed effects model. Swimming effort increased with current speed in northbound current, but this connection was not present in southbound current (t = -2.358, df = 440, P = 0.019, Fig 7). Wave length interacted with current speed in such a way that in weak current, salmon had slower tail beats in longer waves, but in stronger currents, they had faster tail beats in longer waves (t = 2.309, DF = 440, P = 0.021, Fig 7).

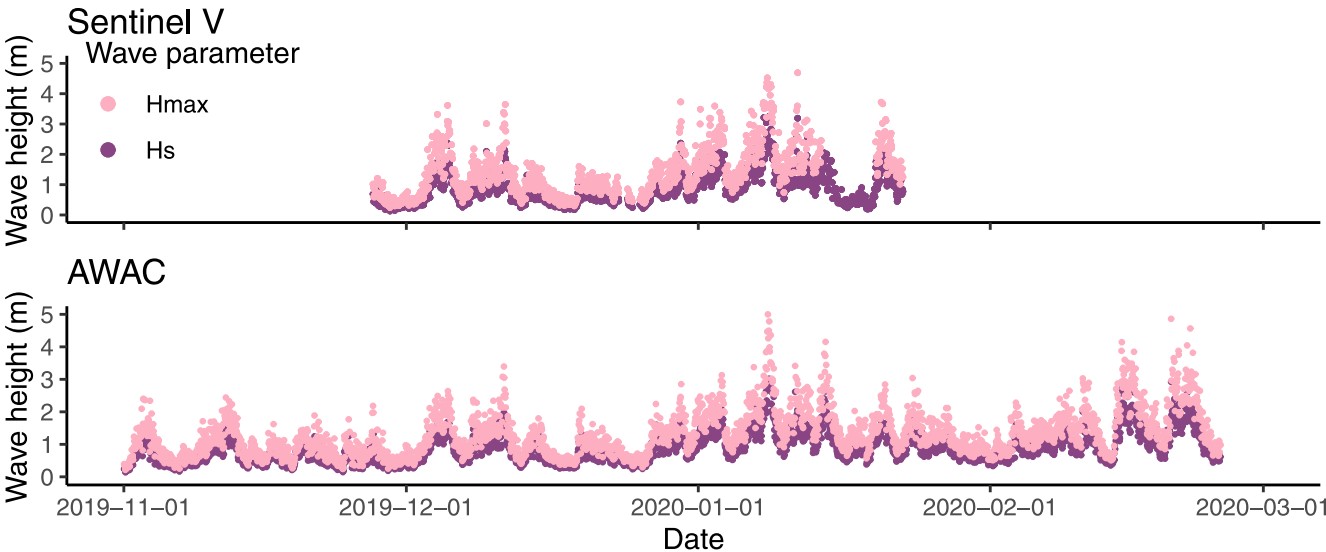

**Fig 4. Wave height measured by two current profilers.** The wave heights measured throughout the monitoring period. The Sentinel V was located south of the study cage whereas the AWAC was located north of the farm.

**Cage perimeter interactions.** Salmon showed great horizontal preference ($F_{4,160}$ = 8.641, P < 0.001) with the "South" camera recording "Many" salmon 36% of the time compared to the "East" camera, which only recorded "Many" salmon 4.5% of the time (Fig 8). The amount of time where "Many" salmon were recorded in cameras decreased overall in large waves (combined high Hm0 and Tp) ($F_{3,161}$ = 3.398, P = 0.019, residual df = 159), though this change varied by camera with the "North" and "South" cameras showing little or even the opposite change (Fig 8). However, in taller waves the total amount of time where any salmon were

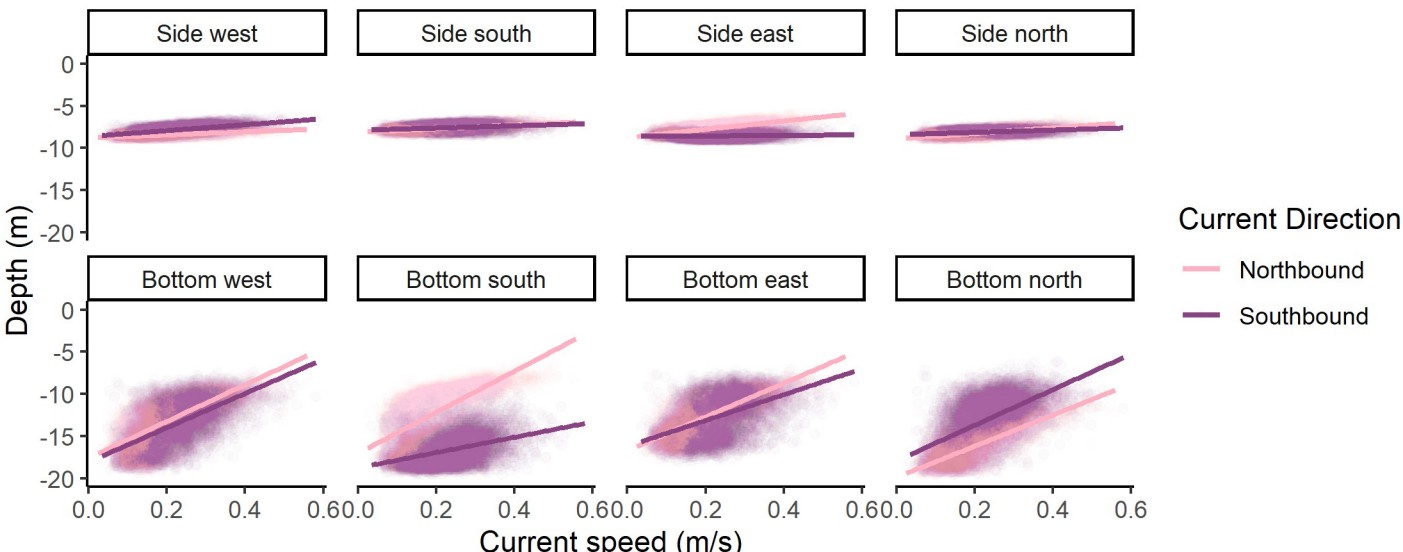

**Fig 5. Depth measured by pressure sensors.** Depth measured by pressure sensors located at the bottom of the side net (Side- west, south, east, and north) and half way from the side to the centre (Bottom- west, south, east, and north). Points are depths measured and lines are lines of best fit. Darker points and lines are depths in southbound current and light points and lines are depths in northbound current.

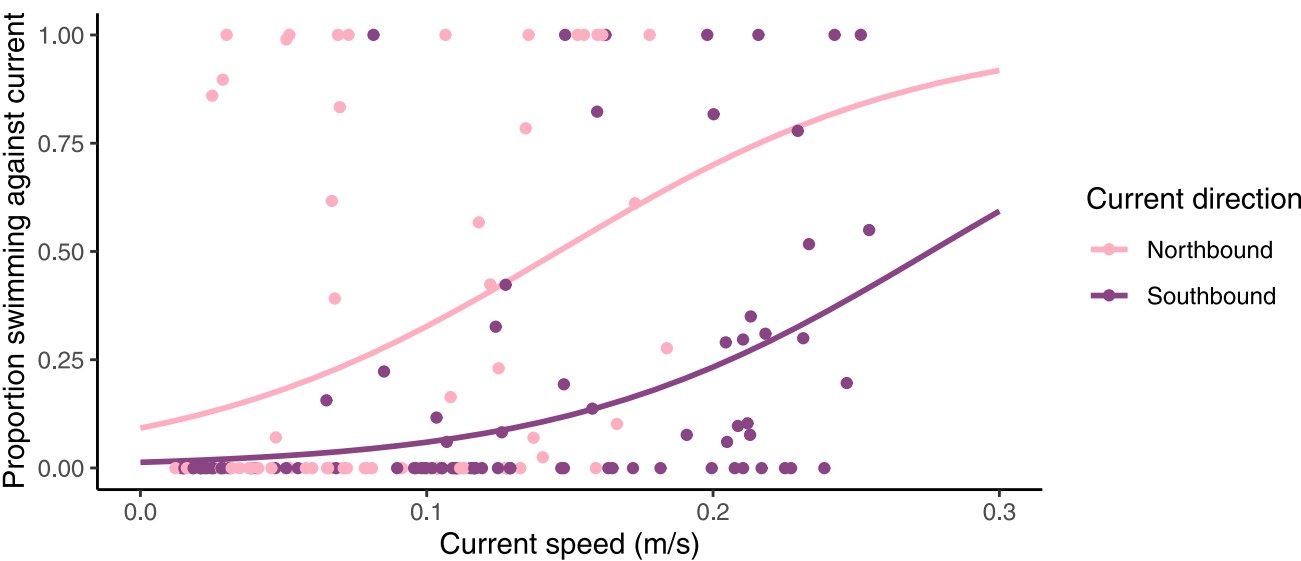

**Fig 6. Proportion of time spent by the majority of the salmon in frame standing on current.** Points are proportion of time where the salmon were standing on current in a video. Lines represent model predictions of the probability of a video having salmon standing on current a majority of the time. Model predictors are current speed and direction. Lighter colour signifies a northbound current direction whereas a darker colour signifies a southbound current direction.

observed in the "East" camera increased, with the proportion of time when any ("Some", "Many", or "Shoal") salmon were visible in the "East" camera increasing from 50% in waves up to and including 1.1 m tall to approximately 75% in waves taller than 1.1 m ($F_{4,155}$ = 7.545, P < 0.001).

Collisions with the net decreased in taller waves, but less so if current was strong too (z = 2.431, DF = 41, P < 0.015, Fig 9).

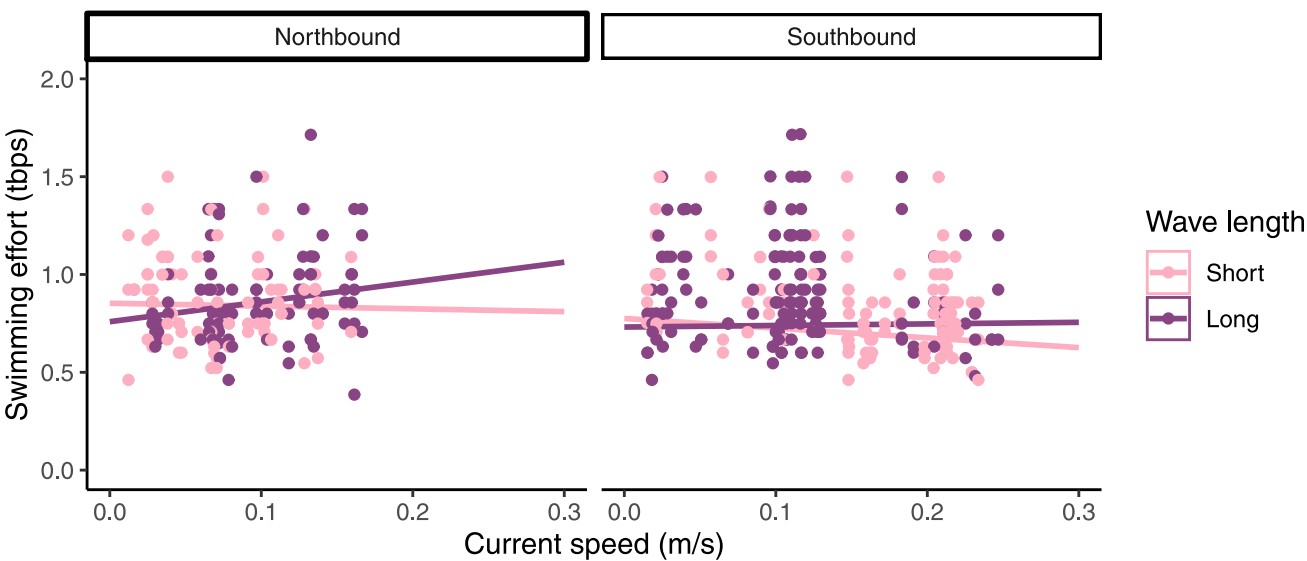

**Fig 7. Swimming effort measured in tail beats per second (tbps) over current.** Swimming effort in salmon standing on current in northbound and southbound currents. Shading indicates wave length less than 12 seconds (lighter colour) and more than 12 seconds (darker colour).

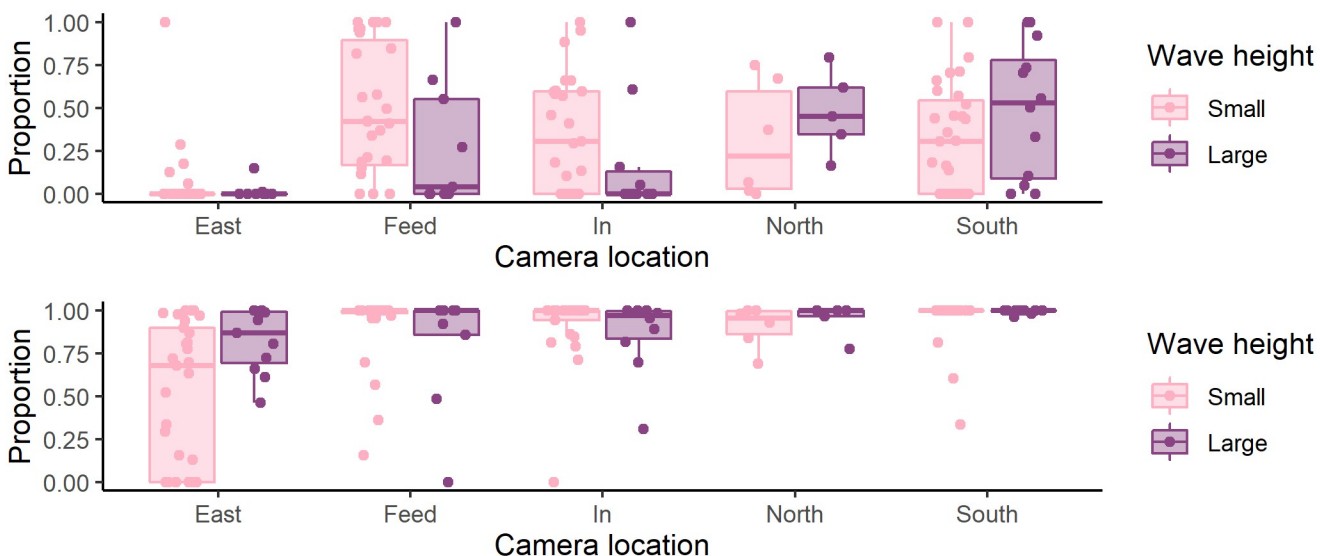

**Fig 8. Proportion of time per video where "Many" (top panel) or "Any" (bottom panel) salmon were recorded.** Light boxes indicate small waves and dark boxes indicate large waves. Dots are raw data and boxes and whiskers represent quartiles. Wave height was split at 1.1 m with "Small" waves being up to and including 1.1 m tall and "Large" waves being taller than 1.1 m. This was an approximate 50/50 split of the available data, and does not necessarily represent biological significance. Statistics are carried out using continuous wave parameters.

## Shoal cohesion and position

Salmon were less likely to shoal in larger waves ($z = -2.023$, DF = 74, $P = 0.043$) and when the current was southbound ($z = -2.185$, DF = 74, $P = 0.029$). The occurrence of shoaling differed between cameras with no shoaling observed in the "East", "North", and "Feed" cameras and

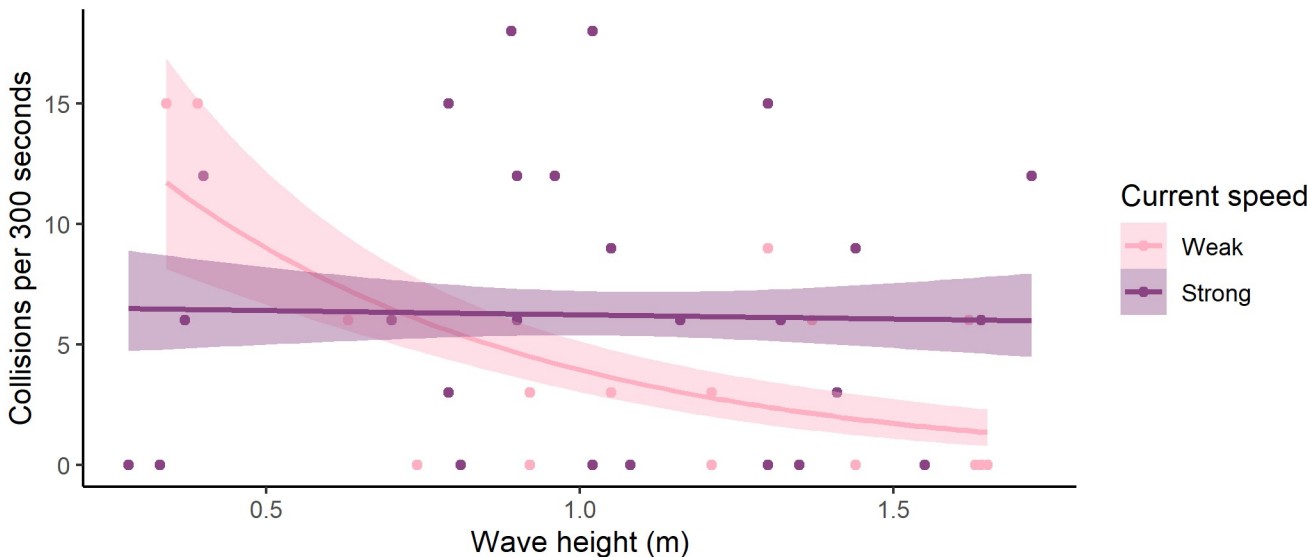

**Fig 9. Number of net collisions per 300 seconds over wave height split by current speed.** Dots are number of collisions normalised to per 300 seconds of video containing salmon over measured wave height. Lines are best lines of fit using a poisson fit. Current was split at 0.2 m s$^{-1}$ with light dots and lines representing currents up to and including 0.2 m s$^{-1}$ and dark lines and dots representing currents stronger than 0.2 m s$^{-1}$.

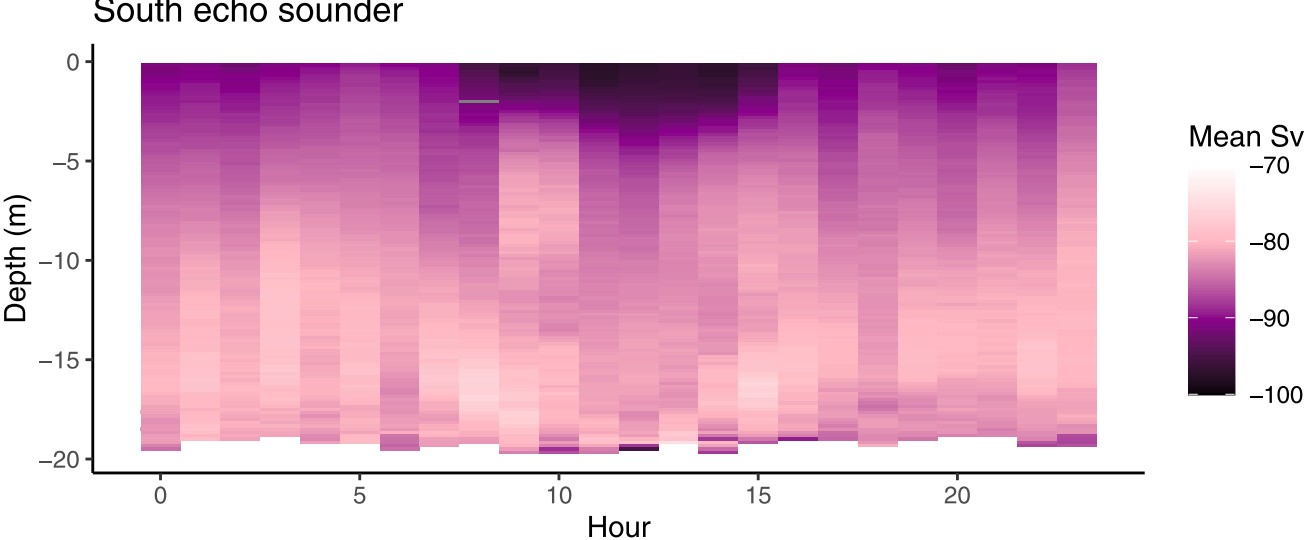

**Fig 10. Depth and dispersal of fish seen in the south echo sounder over hour in the day when current is weak.** Back scatter strength ($S_v$) indicates fish density, with higher values (-70) indicating many fish and lower values (-100) indicating fewer fish. Data presenting current stronger than 0.11 m s$^{-1}$ is not included for simplicity.

with more shoaling observed in the "South" camera compared to the "In" camera (z = 2.46, DF = 74, P = 0.014).

Fish avoided the surface during the day (Fig 10) and this pattern persisted regardless of hydrodynamic conditions.

As current increased, the fish narrowed their vertical distribution within the water column resulting in stronger localised $S_v$ (South echo sounder; $F_{3,250864}$ = 2993, P < 0.001, North echo sounder; $F_{3,239438}$ = 1836, P < 0.001) and greater residuals (South echo sounder; $F_{3,250864}$ = 759.1, P < 0.001, North echo sounder; $F_{3,239438}$ = 1265, P < 0.001) as opposed to a lower signal with more even dispersal in weaker currents (Fig 11).

Fish moved upwards in the water column in stronger current. The side of the cage and direction of current affected the degree to which the shoals moved up (South echo sounder; $F_{3,218622}$ = 8665, P < 0.001, North echo sounder; $F_{3,206399}$ = 7170, P < 0.001, Fig 11). They still avoided the surface during the day, resulting in a greater concentration of fish below 5m depth (Fig 11).

In an effect similar to that of daylight, waves caused fish to move away from the surface (South echo sounder; $F_{4,218621}$ = 6307, P < 0.001, North echo sounder; $F_{3,206399}$ = 6026, P < 0.001). However, at the side of the cage where current entered the cage, fish moved upward, countering the effect of waves (Fig 12).

## Welfare

While injury scores were low throughout the entire production cycle with 90% of salmon scoring three or lower in injuries most of the time, there is significant variation between sampling times with a period in January and February where more than 25% of the salmon have a score of four or higher (Chi squared = 168.62, DF = 16, P < 0.001, Fig 13). In late June, when the salmon were harvested, higher scoring fish had decreased again to less than 20% (Fig 13). Throughout the entire observation period, only one fish scored more than 6 and only during the harshest winter months were a substantial number of salmon scoring 5 and 6 found.

## North echosounder

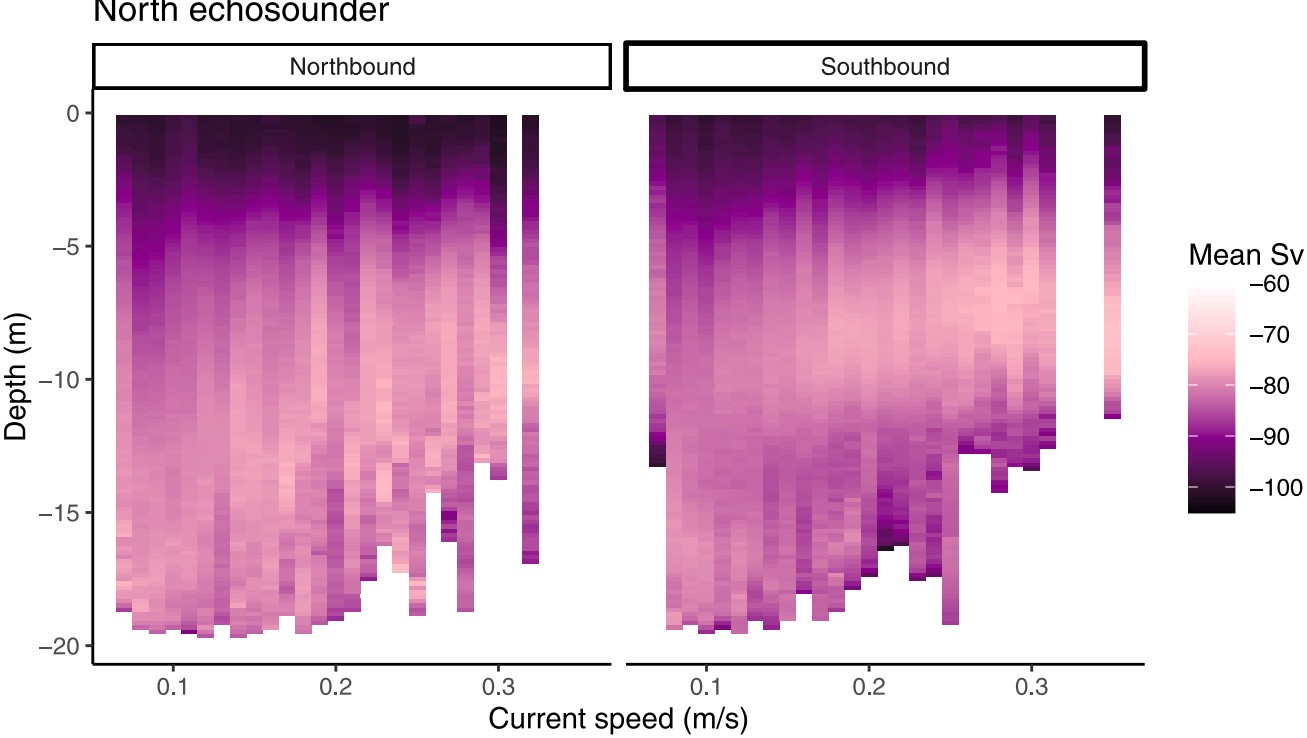

**Fig 11. Depth and dispersal of fish over current.** The upper and lower panels indicate the two echo sounders, and the left and right panels the direction of current. Back scatter strength ($S_v$) indicates fish density, with higher values (-70) indicating many fish and lower values (-100) indicating fewer fish. Data presenting shoal depth during the night is not included for simplicity.

## Discussion

At the study site, currents caused the cage to deform, which is consistent with measurements elsewhere indicating a minimum volume loss of 20% at 0.5 m s$^{-1}$ [24, 25]. The bottom of the salmon cage moved upwards on the side of incoming current from almost 20 metres depth up to almost 5 metres in extremes. This means that in stronger currents, salmon are unable to use

## North echosounder

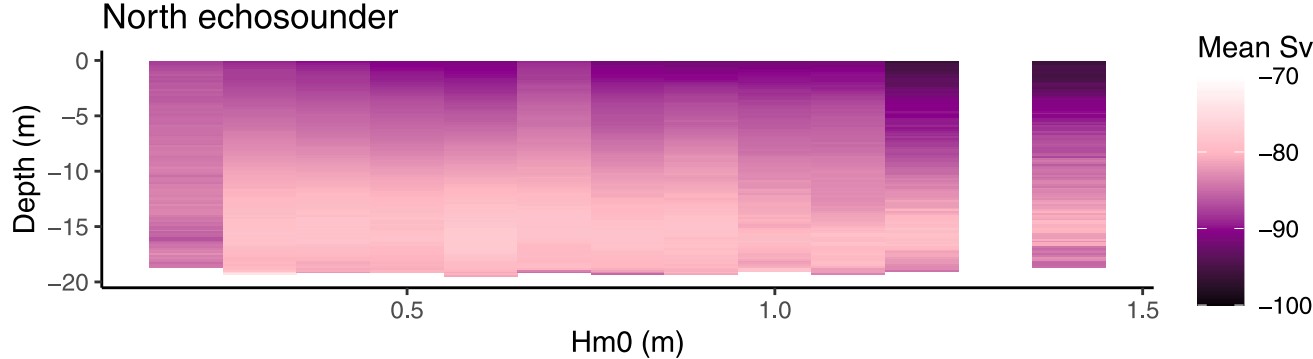

**Fig 12. Depth and dispersal of fish over wave height (Hm0) in weak current.** The left and right panels are the south and north echo sounder. Data where current exceeded 0.11 m s$^{-1}$ are excluded. Signal strength ($S_v$) indicates fish density with higher values (-70) indicating many fish and lower values (-100) indicating fewer fish. Data presenting shoal depth during the day not included for simplicity.

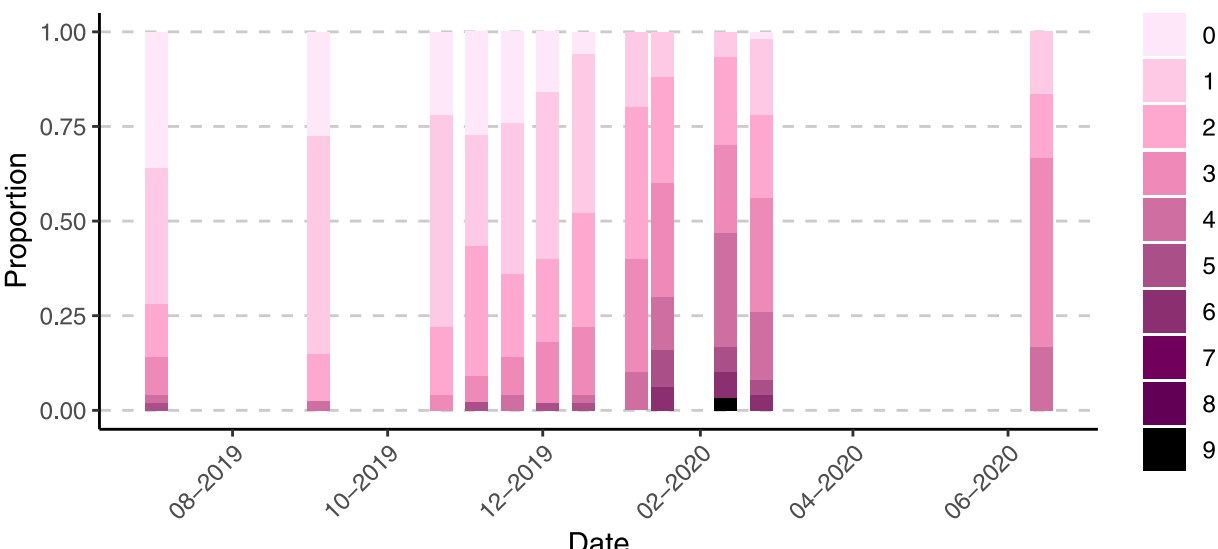

**Fig 13. Distribution of welfare scores over time.** Proportion of salmon with summed welfare scores between 0 and 9. A score of zero indicates that the fish had no injury at all and higher scores are more injuries.

the cone portion of the cage on the side where the current enters the cage, and even in the sheltered side, the cone moves upwards in very strong currents.

Because of surface avoidance during daylight hours [54], salmon at this study site preferred to occupy the cone portion of the cage during the day, while they occupied the entire water column during the night. This caused a concentration of fish above the cone (upwards of 10 m) in stronger currents, because a major component of cage deformation is the bottom of the cage being pushed upwards [25]. When strong currents peaked during daylight hours, this caused a greater concentration of fish in the 5-10 m depth range above the cone and away from the surface. In our previous paper [27], it was not clear whether the observed change in vertical position caused by current was related to cage deformation or a choice by the salmon. Here, we show that the bottom of the shoal does move upwards almost exactly as the bottom of the cage. Therefore, there is no evidence to suggest that the salmon move away from the bottom of the cage at any time, rather that they adjust to the change in space available to them.

In addition to the effect of daylight and current, waves also affected vertical shoal positioning. At night, salmon avoided the surface in tall waves similarly to how they respond to daylight. Due to the effect of cage deformation, this preference was not so clear on the side of the cage affected by oncoming current. Whether this is caused by an interaction between current and waves, or simply that surface aversion due to daylight is stronger than that caused by waves is uncertain. However, daylight combined with large waves caused fish to move further down as evidenced by both echo sounder data and video data. This corresponds well with our previous paper [27] as well as what was seen in Signar P. Dam's study [28]. Very little other work has been done directly on salmon and waves in farming, but there is some theoretical work indicating that in long period waves out at sea, the horizontal movement of the waves may exceed the swimming capacity of salmon [55], particularly if the cages are not deep enough for the salmon to dive below the waves.

In terms of association with the net, there was some indication that there were fewer salmon near the net in waves that were both tall and long, but the most decisive change was in the more sheltered part of the cage, where the median proportion of time where salmon were seen

in the video increased by 50 percentage points. This indicates that there was more dispersal of salmon from the more popular exposed side of the cage, to the more sheltered side when the waves were large. Despite little to no change in salmon seen in the "North" and "South" cameras and the increase of salmon seen in the "East" camera, collisions with the net generally decreased as wave height increased, but only in weak current. It is possible that strong current negatively affects salmon's ability to adjust swimming speed to accommodate waves, or this difference might be due to a greater proportion of salmon standing on current near the net in strong current. Free swimming salmon do not necessarily associate as strongly with the net.

As expected, salmon swimming mode was affected by current speed with fish changing from swimming freely to standing on current as current increased [23]. Perhaps unintuitively, the salmon beat slower with their tails once standing on current. Literature suggests that salmon change from circular swimming to standing on current when the current exceeds their preferred swimming speed [56]. However, at this location, salmon changed swimming mode at fairly low currents, so the swimming effort was actually lower at that current speed than their cruising speed. While there were some effects of waves and currents on swimming effort, neither had a major effect, and the low speed at which the salmon switched to standing on current might indicate that they prefer to do so at this site. Literature suggests that forcing salmon to exceed their preferred swimming speed might have a negative influence on salmon [57], but considering that current speeds rarely reached one body length per second (1 BL s$^{-1}$) at this site, this is unlikely to be a problem for these salmon, and might in fact perform the role of environmental enrichment instead [58, 59]. Considering that salmon were also seen less in the "East" camera, there is clear indication that even at low currents, salmon in this cage do not perform a circular swimming pattern using the entire perimeter of the cage as is seen in other farms [23, 60]. While currents were weak by 50 cm long salmon standards, these cages were also inhabited by lumpfish, and it is possible that the strongest currents at this site exceed lumpfish swimming capacity [61]. Lumpfish welfare data were not collected for this study. However, the farmers at the site were aware of potential implications of currents, and had deployed shelters that were adapted for use in strong currents, providing firm surfaces to attach to as well as shelter from the current. As opposed to our previous paper [27], swimming effort did not decrease in larger waves, but rather increased (though only in long period waves). The effect of wave period was most apparent in stronger currents, which were not present at the previous study site. It may also be that the larger waves seen in the video footage in that study were tall waves, but with a shorter period than what was seen in this study. While wave period and height do correlate, this correlation is not perfect, so that too could account for the differences seen.

Taken together, the results indicate that salmon prefer to use the entire water column available to them, and only move upwards in the water column due to cage deformation and downwards due to waves and daylight. However, the data suggest that they do not use all of the available horizontal space. There are a few potential explanations for this. Two potential candidates are; 1) water quality is undesirable in the more sheltered areas of the cage [15, 29], 2) salmon prefer to occupy a space with current. We have no reason to suspect poor water quality in the sheltered parts of the cage, particularly as this study was carried out in the winter months with a lot of mixing in the water column and good general water exchange, even in the sheltered areas. Therefore, the more likely explanation is that the salmon actively choose the more exposed side of the cage rather than avoid the sheltered side. We have found no literature investigating preferential swimming in currents in a salmon cage other than in relation to water quality parameters, but it appears as though there is preference for current regardless of water quality.

OWIs remained low for the majority of fish throughout the production cycle. Welfare indicator scores were slightly elevated in January and February of 2020 directly after large wave events. However, fewer collisions were seen in larger waves, so if these elevated OWIs were caused by collisions, they must have happened off-camera. There are three reasons this could be: 1) collisions happened at night, 2) collisions happened elsewhere in the cage, 3) power cut out during bad weather preventing us from seeing the collisions in the largest waves. That being said, the OWIs were only slightly elevated during these months and there are numerous other explanatory variables, which we were unable to control for. Therefore, one ought not necessarily conclude that bad weather events caused poor welfare.

This study is limited by the available study locations. Particularly that only one cage was monitored for salmon behaviour. This limits our conclusion to this one cage, but considering that the current literature on salmon in commercial cages is also often limited to one or a few units, these data add to what is already a limited pool of information.

## Conclusion

The strongest effects that currents and waves have on salmon relate to how they decrease the space available to the fish. Salmon actively choose to occupy areas in the cage exposed to stronger currents, but avoid the surface in large waves. For farming in new sites exposed to strong currents and potentially also in currently used sites, one ought to consider the limitation to vertical space when currents cause cage deformation. Therefore, making the cage resistant to deformation and deep enough that even in the strongest currents, there is enough vertical space, can be beneficial to salmon welfare. This is particularly relevant during the day, when waves are large, and if the biomass is high, for example close to harvest. At sea, waves are likely to have longer periods, necessitating a deeper diving depth to avoid them. Alternatively, salmon will have to move away from the sides of the cage to avoid collision, which effectively reduces the diameter of the cage, greatly reducing the volume.

## Acknowledgments

We would like to thank Hiddenfjord for cooperating with us in this project. Without access to their farm and the help of their staff on site, this work would not be possible.

## Author Contributions

**Conceptualization:** Ása Johannesen, Øystein Patursson, Signar Pætursonur Dam, Pascal Klebert.

**Data curation:** Ása Johannesen.

**Formal analysis:** Ása Johannesen, Jóhannus Kristmundsson.

**Funding acquisition:** Øystein Patursson, Pascal Klebert.

**Investigation:** Ása Johannesen, Øystein Patursson, Jóhannus Kristmundsson, Mats Mulelid, Pascal Klebert.

**Methodology:** Ása Johannesen, Øystein Patursson, Jóhannus Kristmundsson, Signar Pætursonur Dam, Pascal Klebert.

**Project administration:** Ása Johannesen, Øystein Patursson, Pascal Klebert.

**Resources:** Øystein Patursson, Signar Pætursonur Dam, Pascal Klebert.

**Software:** Jóhannus Kristmundsson.

**Supervision:** Øystein Patursson, Pascal Klebert.

**Visualization:** Ása Johannesen.

**Writing – original draft:** Ása Johannesen.

**Writing – review & editing:** Ása Johannesen, Øystein Patursson, Jóhannus Kristmundsson, Signar Pætursonur Dam, Mats Mulelid, Pascal Klebert.

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
