## [Decision Letter · Decision Letter 0]

23 Dec 2021

PONE-D-21-33503Waves and currents decrease the available space in a salmon cagePLOS ONE

Dear Dr. Johannesen,

Thank you for submitting your manuscript to PLOS ONE. After careful consideration, we feel that it has merit but does not fully meet PLOS ONE’s publication criteria as it currently stands. Therefore, we invite you to submit a revised version of the manuscript that addresses the points raised during the review process.

I agree with both referee that the review work done on the manuscript was very effective. Only one of the reviewers suggested a minor revision (see the report).

We look forward to receiving your revised manuscript.

Kind regards,

Pierluigi Carbonara, PhD

Academic Editor

PLOS ONE

Journal Requirements:

Reviewers' comments:

Reviewer's Responses to Questions

**Comments to the Author**

1. Is the manuscript technically sound, and do the data support the conclusions?

Reviewer #1: Yes

Reviewer #2: Yes

2. Has the statistical analysis been performed appropriately and rigorously? 

Reviewer #1: Yes

Reviewer #2: Yes

3. Have the authors made all data underlying the findings in their manuscript fully available?

Reviewer #1: Yes

Reviewer #2: Yes

4. Is the manuscript presented in an intelligible fashion and written in standard English?

Reviewer #1: Yes

Reviewer #2: Yes

5. Review Comments to the Author

Reviewer #1: In the previous version of the manuscript, I had some major concerns, mainly about missing details concerning methods and analysis, as well an insufficient discussion of the results. In the revised version, Ása Johannesen et al. did a careful revision of the manuscript, answering all of my concerns and those of the second reviewer. I want to congratulate them for their work, and so my recommendation is to accept the manuscript.

Minor comment:

Ln 415 : please, could you provide a reference to support the statement that such current spped might actually perform the role of environmental enrichment.

Reviewer #2: The authors have followed the suggestions in a very satisfactory way, and I therefore recommend this work for publication. The authors have followed the suggestions in a very satisfactory way, and I therefore recommend this work for publication.

6. PLOS authors have the option to publish the peer review history of their article (what does this mean?). If published, this will include your full peer review and any attached files.

Reviewer #1: **Yes: **Sébastien Alfonso

Reviewer #2: No

---

## [Author Response · Author response to Decision Letter 0]

18 Jan 2022

Reviewer 1 had one comment:

Ln 415 : please, could you provide a reference to support the statement that such current spped might actually perform the role of environmental enrichment.

We have added references to two papers detailing how current speed may be considered environmental enrichment. Unfortunately, it hasn't been that well documented for uses other than promoting growth and decreasing aggression. How currents affect salmon's internal state is not so clear and ought to be better investigated.

---

## [Editor Report · Decision Letter 1]

28 Jan 2022

Waves and currents decrease the available space in a salmon cage

PONE-D-21-33503R1

Dear Dr. Johannesen,

We’re pleased to inform you that your manuscript has been judged scientifically suitable for publication and will be formally accepted for publication once it meets all outstanding technical requirements.

Kind regards,

Pierluigi Carbonara, PhD

Academic Editor

PLOS ONE

---

## [Editor Report · Acceptance letter]

8 Feb 2022

PONE-D-21-33503R1 

Waves and currents decrease the available space in a salmon cage 

Dear Dr. Johannesen:

I'm pleased to inform you that your manuscript has been deemed suitable for publication in PLOS ONE. Congratulations! Your manuscript is now with our production department. 

Kind regards, 

on behalf of

Dr. Pierluigi Carbonara 

Academic Editor

PLOS ONE